# Gut Microbiota and Immune System Interactions

**DOI:** 10.3390/microorganisms8101587

**Published:** 2020-10-15

**Authors:** Ji Youn Yoo, Maureen Groer, Samia Valeria Ozorio Dutra, Anujit Sarkar, Daniel Ian McSkimming

**Affiliations:** 1College of Nursing, University of South Florida, Tampa, FL 33612, USA; mgroer@usf.edu (M.G.); anujit@usf.edu (A.S.); 2College of Nursing, University of Tennessee-Knoxville, Knoxville, TN 37916, USA; sozoriod@utk.edu; 3College of Public Health, University of South Florida, Tampa, FL 33612, USA; 4Morsani College of Medicine, University of South Florida, Tampa, FL 33612, USA; dmcskimming@usf.edu

**Keywords:** gut microbiota, immune system, gut microbiota metabolites, short-chain fatty acids (SCFAs), gut dysbiosis

## Abstract

Dynamic interactions between gut microbiota and a host’s innate and adaptive immune systems play key roles in maintaining intestinal homeostasis and inhibiting inflammation. The gut microbiota metabolizes proteins and complex carbohydrates, synthesize vitamins, and produce an enormous number of metabolic products that can mediate cross-talk between gut epithelial and immune cells. As a defense mechanism, gut epithelial cells produce a mucosal barrier to segregate microbiota from host immune cells and reduce intestinal permeability. An impaired interaction between gut microbiota and the mucosal immune system can lead to an increased abundance of potentially pathogenic gram-negative bacteria and their associated metabolic changes, disrupting the epithelial barrier and increasing susceptibility to infections. Gut dysbiosis, or negative alterations in gut microbial composition, can also dysregulate immune responses, causing inflammation, oxidative stress, and insulin resistance. Over time, chronic dysbiosis and the translocation of bacteria and their metabolic products across the mucosal barrier may increase prevalence of type 2 diabetes, cardiovascular disease, inflammatory bowel disease, autoimmune disease, and a variety of cancers. In this paper, we highlight the pivotal role gut microbiota and their metabolites (short-chain fatty acids (SCFAs)) play in mucosal immunity.

## 1. Gut Microbiota Metabolites (SCFAs)

Understanding interactions between a host immune system and the tens of trillions of microbes that live in a human’s gastrointestinal (GI) tract known as the gut microbiota is an active area of research [1]. When operating optimally and under normal circumstances, the alliance between the immune system and gut microbiota interweaves the innate and adaptive arms of immunity in a dialog that selects, adjusts, and terminates responses in the most appropriate manner [2]. Gut microbial metabolites play key roles in inflammatory signaling, interacting both directly and indirectly with host immune cells [1]. Some bacteria, including *Faecalibacterium prausnitzii*, *Roseburia intestinalis*, and *Anaerostipes butyraticus* [3] digest complex carbohydrates via fermentation, creating short-chain fatty acids (SCFAs) [4,5,6] that modulate host immune cells and serve as a carbon source for colonocytes [7,8]. SCFAs are fatty acids with fewer than six carbon atoms, mainly consisting of acetate, propionate, and butyrate [1,9]. Collectively, SCFAs are considered the most abundant microbiota-derived metabolites in the gut lumen and are enriched by their vigorous capacity to reduce intestinal inflammation, protect against pathogen invasion, and maintain barrier integrity largely by activating G-protein coupled receptors (GPCRs) or inducing their suppressive effects on histone deacetylases (HDACs), influencing gene expression.

Colonocytes absorb SCFAs, especially butyrate, via sodium-dependent monocarboxylate transporter-1 (SLC5A8). SLC5A8-mediated entry of butyrate from the lumen into colonic epithelial cells leads to inhibition of HDAC, epigenetically altering gene expression. SCFAs also activate G protein-coupled receptors (GPRs), including GPR41, GPR43 (also known as free fatty acid receptors (FFAR)-3 and -2), GPR109a (also known as HCA2, niacin/butyrate receptor) and olfactory receptor-78 (Olfr-78) [10,11,12]. SCFAs have various functions depending on the tissue or cell type involved. For example, SCFAs maintain intestinal epithelium physiology by regulating cellular turnover and barrier functions. SCFAs are also a crucial regulator for the activation, recruitment, and differentiation of immune cells, including neutrophils, macrophages, dendritic cells (DCs), and T-lymphocytes. SCFAs can have anti-inflammatory effects on host immune cells, regulating the expression of pro-inflammatory cytokines including Interleukin-6 (IL-6), Interleukin-12 (IL-12), and tumor necrosis factor-α (TNF-α) through activation of macrophages and DCs [13]. 

Several studies have demonstrated that butyrate induces the differentiation of T-regulatory (T_reg_) cells in vitro and in vivo, and ameliorates the development of colitis induced by the adoptive transfer of CD4^+^CD45RB^hi^ T-cells in Rag1−/− mice [14]. Butyrate also modulates the differentiation of anti-inflammatory forkhead box protein P3 (Foxp3), which has an essential role in the suppression of inflammatory responses [14,15]. Moreover, butyrate controls the cytokine production profile of T-helper (T_h_) cells and promotes intestinal epithelial barrier integrity, which can help limit exposure of the mucosal immune system to luminal microbes and prevent aberrant inflammatory responses [16]. By binding GPR109a on DCs and macrophages, the presence of butyrate leads to increased expression of IL-10 and decreased production of IL-6, resulting in increased T_reg_ cell development while inhibiting the expansion of pro-inflammatory Th17 cells. Therefore, GPR109a potentiates anti-inflammatory pathways, induces apoptosis, and protects against inflammation-induced colon cancer [11]. However, GPR109a activation in keratinocytes induces flushing by activation of Cox-2-dependent inflammatory signaling, with receptor expression commonly upregulated in human epidermoid carcinoma [17]. Depending on the cellular context and tissue, GPR109a activation can act as either tumor suppressor or promoter.

Acetate, an SCFA highly produced by *Bifidobacteria* species, also regulates intestinal inflammation by stimulation of GPR43, [18] helping maintain gut epithelial barrier function [19]. Maslowski and colleagues [18] found that GPR43 activation by SCFAs was necessary for the normal resolution of some inflammatory responses, with GPR43-deficient (Gpr43−/−) mice showing an inability to resolve inflammation in models of colitis, arthritis, and asthma. In addition to demonstrating increased production of inflammatory mediators and increased immune cell recruitment by Gpr43−/− immune cells, Maslowski et al. also demonstrated a similar dysregulation of inflammatory responses in germ-free mice, strongly suggesting GPR43 binding of SCFAs as a molecular link between gut bacterial metabolism and host immune and inflammatory responses. Moreover, SCFA-mediated GPR43 signaling attenuates NOD-, LRR- and pyrin domain-containing protein 3 (NLRP3) inflammasome activation and subsequent secretion of IL-18, [20,21], a pro-inflammatory cytokine with essential roles in colonic inflammation and inflammation-associated cancers [11,22]. Acetate also plays an anti-inflammatory role in neutrophils by reducing NF-kB activation via inhibiting the expression of pro-inflammatory mediators such as lipopolysaccharide-induced TNF-α, though to a lesser degree than propionate or butyrate [23,24].

SCFAs have multiple mechanisms of inhibiting inflammation in the gut [25]. SCFAs are well-known HDAC inhibitors, stimulating histone acetyltransferase activity and stabilizing hypoxia-inducible factors (HIFs). HDACs are enzymes that cleave acetyl groups from acetyl-lysine amino acid in histones and various non-histone proteins to change the conformation of the nucleosome, thereby regulating gene expression. SCFA-driven inhibition of HDACs tends to produce immunological tolerance, an anti-inflammatory cell phenotype that is crucial for maintaining immune homeostasis. HDAC inhibitors can directly inhibit tumor cell proliferation as a result of inducing cell cycle arrest and apoptosis through epigenetic effects on gene expression and by interfering with T-cell chemotaxis in the tumor microenvironment [26]. Many studies have observed that HDACs are inhibited by SCFAs, mainly propionate and butyrate, leading to suppression of inflammatory responses in immune cells and tumors [13,24,26,27,28]. The inhibition of HDACs is the most likely mechanism underlying the blockade of DC development by butyrate and propionate [11]. Exposure of peripheral blood mononuclear cells and neutrophils to SCFAs, similar to their exposure to global HDAC inhibitors, has been demonstrated to inactivate NF-κB and downregulate production of the pro-inflammatory cytokine and TNF-α [29]. Butyrate inhibits HDACs via induced Zn^2+^ binding in their active site [30], boosting the acetylation of histone H3 at the Foxp3 promoter and at enhancer-conserved noncoding sequences (CNSs), eliciting robust gene expression and functional maturation [31,32]. 

## 2. Interactions between Gut Microbiota and Immune Cells

### 2.1. Immune Regulation of Gut Microbiota

Intestinal epithelial cells (IECs) are an integral component of the innate immune system and affect the intestinal microenvironment through the identification and uptake of SCFAs, using both passive and active mechanisms. IECs metabolize most of the absorbed butyrate, while propionate is largely taken up by the liver and acetate reaches the systemic circulation at higher concentrations. Butyrate derived from commensal bacteria induces the production of transforming growth factor β (TGF-β) in IECs, a process mediated by its HDAC inhibitory activity and through transcription factor-specific protein binding on the core promoter, driving TGF-β1 expression in IECs and the subsequent convergence of T_reg_ cells in the gut [33,34]. 

The maintenance of mucosal immunologic homeostasis is an enormous task demanding discrimination between billions of beneficial microbes and rare, pathogenic invaders. Gut homeostasis is characterized by the dominance of obligate anaerobic members of *Firmicutes* and *Bifidobacteriaceae*, whereas an expansion of facultative anaerobic *Enterobacteriaceae* is a common marker of gut dysbiosis [35]. Under gut homeostatic conditions, peroxisome proliferator-activated receptor gamma (PPAR-γ, a nuclear receptor primarily synthesized in IECs) is activated by butyrate. PPAR-γ promotes the mitochondrial β-oxidation of SCFAs as well as oxidative phosphorylation in colonocytes, thereby maintaining a local hypoxic microenvironment [35]. The obligate anaerobic SCFA-producing bacteria grow vigorously in such an environment, while the facultative anaerobic enteric pathogens’ growth is suppressed [35,36]. Simultaneously, PPAR-γ activation suppresses NOS2 expression in IECs, as well as the production of inducible NO synthase and nitrate, an essential energy source for facultative anaerobic pathogens [35]. Moreover, propionate has been shown to confer colonization resistance to pathogens in a PPAR-γ-independent manner, suggesting functional redundancy present in SCFAs. Indeed, SCFAs mediate the intracellular acidification of pathogens, which is protective against pathogen infection. For example, one important function of propionate is to limit pathogen expansion via facilitating the cytoplasmic acidification of *Salmonella* or *Shigella*, disrupting the intracellular pH homeostasis of the pathogens. Accumulation of SCFAs and the acidic environment reverse or counteract the competitive advantage that O_2_ and NO_3_ respiration provide to facultative anaerobes like *Enterobacteriaceae*. Conversely, inhibition of the PPAR-γ signaling pathway induces metabolic reprogramming, gut dysbiosis, and SCFA exhaustion. This reprogramming shifts colonocytes towards anaerobic glycolysis, called the Warburg effect, and away from oxidative metabolism, which markedly elevates the levels of oxygen, nitrate, and lactate in the gut lumen [35]. Additionally, an abundance of *Enterobacteriaceae,* such as *Salmonella or Shigella,* utilizes virulence factors to induce neutrophil transepithelial migration, contributing to diminished concentrations of SCFAs. This negative feedback loop creates an environment that is more conducive to pathogen growth, and demonstrates a causal interaction between microbiota-derived metabolism and the gut epithelium [37]. SCFAs, particularly butyrate, foster a hypoxic microenvironment by activating PPAR-γ and undermining the pH homeostasis of pathogens to inhibit their colonization (Figure 1). 

Host-body defense mechanisms consist of redundant systems to guard against pathogens. Recognition of gut microbiota initiates with two main pattern recognition receptor systems (PRRs), Toll-like receptors (TLRs) and nucleotide-binding oligomerization domain molecules (NODs) [38,39]. The family of TLRs is the major and most extensively studied class of PRR. PRRs are widely expressed in and on IECs, as well as macrophages and DCs in the intestine. The PRRs recognize microbe- or pathogen-associated molecular patterns (MAMPs or PAMPs) on pathogens and commensals alike [40]. Once a microbe has been recognized, internalized, or has invaded the epithelial layer, it activates an immunologic response appropriate for the microbe. [38]. Upon PAMP recognition, PRRs activate a multitude of intracellular signaling pathways, including adaptor molecules, kinases, and transcription factors, signaling the presence of infection to the host and triggering pro-inflammatory and anti-microbial responses. PRR-induced signal transduction pathways ultimately result in the activation of gene expression and synthesis of a broad range of molecules, including cytokines, chemokines, cell adhesion molecules, and immunoreceptors. The downregulation of pro-inflammatory cytokines, including IL-8, IL-12, and IL-23, mediates protective effects and the upregulation of anti-inflammatory cytokines, such as IL-10, produced by T_reg_ cells [41,42,43]. The DCs present the antigen to naive T-cells and the secretion of anti-inflammatory cytokines initiates propagating systemic and local tolerance [44]. 

Gut microbiota communities differ along the length and width of the GI tract, as well as within the different layers of intestinal mucus. The proximal small intestine is much less active immunologically than the ileum and colon, and the colonization of enteric microorganisms in the small intestine leads to a more permeable intestine that permits the passage of macromolecules and antigens, and may cause immune-mediated pathologic conditions [45]. Gut permeability is closely linked to both commensal microbiota and elements of the mucosal immune system, and is influenced by many factors including gut microbiota modifications, mucus layer alterations, and epithelial damage [46,47]. Gut microbial fermentation products play pivotal roles in host immune responses that maintain mucosal barrier integrity by controlling luminal microbes. For example, TLR-5 recognizes flagellin, the principal component of bacterial flagellum and the structural protein subunit of the flagellar filament. Its TLR-5 agonist activity makes flagellin a dominant antigen for CD4 T-cells and B-cells. This promotes the differentiation of B lymphocytes into IgA-producing cells, which then bind to microbial antigens, neutralizing the activity of the pathogens and preventing infection [48,49]. However, it remains unclear how flagellin crosses the epithelial barrier. 

Commensal bacteria decrease the migration of phagocytes, which traffic microbial antigens to local lymphoid tissues and promote B-cell and T-cell activation. These bacteria stimulate goblet cell differentiation and production of the protective mucosal layer, while pathogenic bacteria induce DCs to secrete pro-inflammatory cytokines [50]. As a result, naive T-cells differentiate into Th1 and Th17 cells, leading to pro-inflammatory immune responses [51]. Different gram-negative bacteria have lipopolysaccharide (LPS) modifications that vary in their potential to stimulate members of the TLRs [52,53]. LPS is a cell wall component characteristic of Gram-negative bacteria that is absent in Gram-positive bacteria. As such, LPS is a representative pathogen-associated molecular pattern that allows mammalian cells to recognize bacterial invasion and trigger innate immune responses. The polysaccharide moiety of LPS primarily plays protective roles for bacteria, such as prevention from complement attacks or camouflage with common host carbohydrate residues. The lipid moiety, termed lipid A, is recognized by the TLR4/MD-2 complex, which transduces signals for activation of host innate immunity. Such modifications are thought to facilitate bacterial evasion of host innate immunity, thereby enhancing pathogenicity. However, the enrichment of SCFA-producing bacteria significantly limits the abundance of Gram-negative bacteria, subsequently decreasing the levels of LPS [54,55]. 

A conspicuous response of the host immune system that follows microbial gut colonization is the production of IgA by the gut-associated lymphoid tissues (GALT). IgA plays a fundamental role in mucosal homeostasis in the gut and functions as the dominant antibody [56]. GALT comprises Peyer’s patches (PPs), interdigitating lymphocytes, plasma cells and lymphocytes present in the lamina propria, and mesenteric lymph nodes. The role of GALT is to manage the immune response via up-take of gut luminal antigens through M-cells, and to initiate antigen-specific immune responses in the host [57]. Studies of germ-free mice have shown several immunodeficiencies, including fewer splenic CD4+ T-cells, structural splenic disorganization, fewer intraepithelial lymphocytes, decreased conversion of follicular-associated epithelium to M-cells, decreased secretory IgA (SIgA), and decreased ability to induce oral tolerance. SIgA functions are the neutralization of bacterial toxins in the gut lumen. For example, during transcytosis (also known as cytopempsis), rotavirus is neutralized through the epithelial barrier, inhibiting the epithelial translocation and inflammatory potential of Shigella LPS. Moreover, IgA binds to M-cells, a cell population present in the epithelium overlying PPs and specialized in the gut luminal across the epithelium [58,59]. SIgA injected into a mouse ligated ileal loop bound selectively to M-cells in a previous study, whereas other immunoglobulins including IgG and IgM did not [60]. Other studies have also demonstrated that the interactions between gut commensal bacteria and mucosal antibody production have been taken up by CD11c+ DCs in the PPs (PP-DCs) that led to the development of the intestinal IgA immune system [61,62]. PPs are secondary immune organs located in the mucosa of the gut. They harbor follicles containing B lymphocytes and intrafollicular T lymphocyte areas. B-cell activation and class-switch recombination from IgM to IgA is supported by PPs and isolated lymphoid follicles [60]. Studies show that SIgA binding to luminal bacteria enhanced their sampling into PPs [62,63]. The absence of IgA homeostatic control results in dysregulation of gut microbiota, which in turn causes hyper-activation of the whole immune system.

### 2.2. Gut Dysbiosis and Immune Dysregulation

An individual’s gut microbiota composition is dynamic, changing in response to age, geographical location, diet, antibiotic use, and influx and efflux of external microbes [64]. Based on their colonization ability, bacteria in the gut can be transient or permanent. Transient bacteria represent microbes that are introduced during adult life from the external environment and do not permanently colonize the intestinal tract for various reasons, such as lack of appropriate adaptations for colonization or inability to compete with the resident microbiota [64]. The health of the microbial community at a site can be ascertained in terms of stability, diversity, resistance and resilience [65]. Briefly, it characterizes the richness of the ecosystem, its vulnerability to compositional and functional change, and its capability of reestablishing itself to its original state. Thus, the balance of the microbial community can be disturbed by either loss of diversity, thriving of pathobionts, or withering of commensals [66]. 

Gut dysbiosis refers to alterations in the composition and function of the gut microbiota that have harmful effects on host health via qualitative and quantitative changes in the intestinal flora itself, changes in their metabolic activities, and/or changes in their local distribution [67]. Some commensal bacteria inhibit the growth of opportunistic pathogens via SCFA production, which alters intestinal pH [19]. For example, *Bifidobacterium* reduces the intestinal pH during fermentation of lactose, thereby preventing the colonization by pathogenic *Escherichia coli* [19,68]. Commensal bacteria do not only inhibit pathogen virulence by changing environmental conditions required for virulence activity, but bacterial metabolites also directly suppress virulence genes of pathogens. For instance, *Shigella flexneri* requires oxygen for the competent secretion of virulence factors, but commensal facultative anaerobes, including members of the *Enterobacteriaceae* family, consume the residual oxygen, leading to incomplete expression of *Shigella* virulence factors in the gut lumen [69]. However, many factors can be a cause of dysbiosis, including invasive intestinal pathogens, antibiotic treatment, physical damage to the mucosa, diet, or host genetic factors [70]. Although the nature of dysbiosis varies according to the individual and pathologic condition, reductions in the relative proportion of obligate anaerobes and increases in facultative anaerobes including pathogens such as *E. coli, Salmonella, Shigella, Proteus*, and *Klebsiella* are common features of dysbiosis [71]. One consequence of dysbiosis is increased susceptibility to enteric infection. If the resident bacterial community is subsequently disrupted by antibiotics treatment, it can induce inflammation. For example, the growth of pathogenic *E. coli* is normally suppressed by commensal bacteria, but antibiotic treatment increases the abundance of *E. coli* in dextran sulfate sodium (DSS)-induced colitis mice, encouraging systemic circulation of the pathogen, thereby promoting activation of the inflammasome [72]. In addition, *Clostridium difficile* (*C. difficile*) is the main cause of the hospital-associated infections, which normally presents at low abundance in the healthy adult gut [73]. Disruption of commensal gut bacteria by broad-spectrum antibiotic treatment in hospitalized patients, lead to a significant increase in the abundance of *C. difficile*, followed by severe gut inflammation [73]. Similar effects have been observed across species, with antibiotic treatment increasing the incidence of *C. difficile* infection in murine models, as well [72]. *C. difficile* produces toxins such as TcdA and TcdB that can destroy the epithelial barrier and increase gut permeability. The toxin-mediated epithelial damage can cause systemic circulation of both bacteria and bacterial metabolites, which is associated with increased inflammation (Figure 2). 

Dysbiosis does not always involve increases in the abundance of pathogens, as the absence of important commensal bacteria can be adverse without the presence of pathogens. Conversely to the outgrowth of potentially pathological bacteria, dysbiosis frequently occurs with diminished bacterial proliferation. Depleted commensals can have important functions, and recovery of the abolished bacteria or their metabolites has the potential to reverse dysbiosis-associated phenotypes [66]. The primary way the host configures gut microbiota is via its immune system. The immune system–gut microbiota crosstalk is of sublime importance in understanding the role of dysbiosis-driven diseases in human. Commensal bacteria prevent pathogen colonization and infection by enhancing the mucosal barrier and promoting innate immune responses. For instance, germ-free and knock-out mice deficient in Toll-like receptor (TLR) signaling adaptor myeloid differentiation primary response protein 88 (MyD88) exhibit distinct intestinal microbiota and diminished production of antimicrobial peptides [74]. Moreover, both increased mucosa-associated bacterial abundance and translocation of bacteria to the mesenteric lymph nodes, in addition to altered bacterial composition, are associated with loss of MyD88 signaling in epithelial cells [75,76]. After infection with pathogenic *Salmonella* or *Pseudomonas,* cytokines like interleukin 1β (IL-1β) are essential for recruitment of neutrophils and elimination of the pathogenic intruders via Nod-like receptors (NLRs) [77]. NLR-containing inflammasomes such as NLRP1, NLRP3, and NLRC4 have been identified [78]. Bacterial toxins stimulate the NLRP3 inflammasome, which drives the proteolytic activation of caspase-1, resulting in the release of mature, biologically active IL-18 and IL-1β (Figure 2) [79]. 

Although members of the microbial community are often considered to be commensals, the relationship can be commensal, mutualistic, or even parasitic. In fact, the interaction between the gut microbe and the immune system can be extremely contextual, defined by the host landscape, diet, and coinfection of the host [2]. It’s a well-known fact that the gut microbiota stimulates the normal development of the immune system and also plays a significant role in the maturation of the immune cells [80]. It has been suggested that gut dysbiosis may induce immune dysregulation and subsequently increase the risk of developing diseases, including inflammatory bowel disease (IBD), diabetes, cardiovascular diseases (CVDs), and autoimmune disease (Figure 3) [81,82].

Diversity and abundance of gut microbiota are emerging as critical determinants of host health, and changes in diversity have been associated with a variety of diseases in humans. It is unclear whether the microbiota participate directly in the pathogenesis of all associated disease states. However, many studies show that gut microbiota contribute directly to the pathogenesis of specific diseases via complex interactions between gut microbiota, host metabolism, and immune systems [83,84,85]. The association between gut dysbiosis and mucosal inflammation is either a cause of dysbiosis, a consequence of it, or some combination of the two, with one study suggesting gut microbiota are essential to the initiation and progression of mucosal inflammation in germ-free mice [86]. One common cause of gut dysbiosis, observed in both clinical and animal models, is infection. But infectious diseases and their treatments also influence the human gut microbiota, creating feedback loops which alter the local environment and ultimately determine the effect of the infection on the host microbes. Numerous studies have verified the intimate relationship between infection and dysbiosis of gut microbiota, and have shown that infection is associated not only with the gut bacteria, but also with resident viruses [87]. For example, *Clostridium difficile* infection patients have gut microbiota that are significantly altered in a manner associated with the progression of the human immunodeficiency virus (HIV), hepatitis B virus (HBV), and other infectious diseases [35,88].

## 3. Dysbiosis of the Gut Microbiota and Related Diseases

The mucosal barrier produced by gut epithelial cells acts as a defense mechanism, segregating microbes from host immune cells and reducing intestinal permeability. Disrupting the epithelial barrier increases susceptibility to infection and the displacement of microbial metabolites into the host. Gut dysbiosis, or negative alterations in the gut microbial composition, not only reduces the integrity of the mucosal barrier, but also dysregulates immune responses, causing inflammation, oxidative stress, and insulin resistance. Over time, chronic gut dysbiosis and the translocation of bacteria and their metabolic products across the mucosal barrier can increase the prevalence of a variety of diseases. Below, we highlight conditions with strong experimental and clinical models linking the subversion of the mucosal immune system and associated environmental responses, like inflammation, with disease onset and severity.

### 3.1. Gut Microbiota and Chronic Low-Grade Inflammation on Type 2 Diabetes (T2DM) and CVD

A number of studies provide evidence supporting associations between gut dysbiosis, T2DM (non-insulin-dependent), and CVD [80,81,82,83]. A major cause of T2DM is impaired insulin action in adipose tissue, skeletal muscle, and the liver [84]. Insulin is a hormone that acts as a key mediator regulating glucose homeostasis and lipid metabolism. Insulin has varied functions, including regulation of gene expression, stimulation of nutrient transport into cells, modification of enzymatic activity, and regulation of energy homeostasis. Insulin also promotes glucose uptake by stimulating translocation of the principal glucose transporter (GLUT4) to the plasma membrane in skeletal muscle and adipose tissue [85]. In skeletal muscle, GLUT4 accounts for up to 75% of insulin-dependent glucose disposal, whereas in adipose tissue it only accounts for a small fraction [86,87]. As a result, impaired skeletal muscle insulin signaling leads to reduced glucose disposal, markedly seen in T2DM patients. Although the exact mechanism in skeletal muscle that leads to the development of insulin resistance is not yet fully understood, increased intramyocellular lipid content and free fatty acid (FFA) metabolites play an essential role in the development of insulin resistance in skeletal muscle [88,89]. Increased circulating FFAs can lead to decreased insulin sensitivity in skeletal muscle due to an increase in intracellular lipid products, including fatty acyl-CoA and ceramide [90,91]. These lipid intermediates can activate the serine/threonine kinase protein kinase C-θ (PKC-θ), which then inhibits insulin signaling. The inhibition of insulin signaling in the liver induces the expression of key gluconeogenic enzymes, resulting in insulin resistance and increased hepatic glucose production [92]. Decreased hormone sensitive lipase activity and the anti-lipolytic effect inhibit FFA efflux from adipocytes by adipose tissue insulin signaling. Insulin resistance or deficiency results in profound dysregulation of these processes and elevates fasting and postprandial glucose and lipid levels, both major causes of T2DM and CVD [93]. 

A well-known gut microbiota related pathway is trimethylamine–N-oxide (TMAO) metabolism. Dietary phospholipids (lecithin, choline, and carnitine) are anaerobically broken down by microbiota to yield the metabolites ethanolamine and trimethyl amine (TMA). Gut microbial TMA lyases can cleave the C-N bond of dietary phospholipids, releasing the TMA moiety as a waste product. These nutrients are all substrates of gut microbiota and dietary precursors of TMA/TMAO production. The conversion from TMA to TMAO requires an oxidation step mediated by host enzyme machinery in the form of flavin monooxygenases (FMOs). Transport via the portal circulation brings the TMA to a cluster of hepatic enzymes, including FMOs that efficiently oxidize TMA, forming TMAO. A study shows that elevated TMAO has been independently associated with prevalence of CVD and incident risks for myocardial infarction (MI), stroke, death, revascularization, and oxidation, and has also been used to predict major adverse cardiac events such as death, MI, and stroke over a three-year period [94]. When carnitine, choline, or betaine circulating levels were elevated and associated with future risk of MI, stroke, or death independent of traditional risk factors, their prognostic values were primarily restricted to those with concomitantly elevated TMAO levels. Wang et al. (2011) [95] also demonstrated that plasma levels of choline, TMAO, and betaine are associated with atherosclerosis risk in humans and promote atherosclerosis in mice. Further studies show TMAO levels are associated with the genera *Prevotella* and *Bacteroides*, and applied dietary TMAO supplementation has promoted a decrease (26% reduced compared to normal chow–fed mice, *p* < 0.01) in total cholesterol absorption in mouse models [96]. 

Numerous studies support the association between chronic low-grade inflammation, T2DM, and CVD [97]. Insulin resistance often results in compensatory hyperinsulinemia, one of the causes of metabolic abnormalities thought to constitute the pathophysiologic basis of metabolic syndrome, a precursor to CVD [87]. Moreover, insulin resistance is related to an excess accumulation of visceral fat, causing chronic low-grade inflammation characterized by increased macrophage infiltration and pro-inflammatory adipokine production. This chronic low-grade inflammation via the inhibition of the insulin-signaling pathway in peripheral tissues is associated with T2DM and the development of CVD [98]. The excess of inflammatory factors such as IL-1, IL-6, and TNF-α are related to the development of impaired insulin action, and are involved in multiple molecular interactions between the immune system and insulin signaling. 

The gut dysbiosis associated with T2DM has been well-documented, including a decrease in the abundance of butyrate-producing bacteria and an increase in various opportunistic pathogens and oxidative stress resistance [99,100]. SCFAs increase anti-inflammatory responses in the host body. However, T2DM patients have shown a significantly lower number of SCFA-producing bacteria and significantly increased LPS levels [97,101]. LPS leakage into the body from gram-negative microbes may be a triggering factor for low-grade inflammation, inducing the release of proinflammatory molecules. Low-grade inflammation is strongly associated with increased triglycerides and decreased HDL cholesterol in the blood, increased blood pressure, and increased fasting plasma glucose [101,102], molecules that increase gut permeability and inflammation in the intestinal epithelium. LPS receptors have been found to be critical mediators that potentially activate underlying insulin resistance. The activation of TLR4 in pancreatic islets was shown to increase proinflammatory cytokine production in macrophages and beta-cells, resulting in decreased function and viability of beta-cells [103]. Moreover, the activation of TLR4 is directly related to decreased mRNA expression of pancreas-duodenum homeobox-1 (PDX-1), insulin gene expression, reduced insulin content, and diminished insulin-induced glucose secretion. Also, LPS upregulates the expression of nuclear factor kappa-light-chain-enhancer of activated B cells (NF-κB) and activates mitogen-activated protein kinase (MAPK)-mediated pathways in adipocytes. The anti-inflammatory effects of SCFAs are probably due to a balance between suppression of proinflammatory mediators and induction of anti-inflammatory cytokines. Butyrate and propionate not only reduce expression of the proinflammatory cytokines TNF-α and IL-6 in human adipose tissue, but also increase the release of the anti-inflammatory cytokine IL-10 by monocytes exposed to bacteria. The ability of SCFAs to reduce low-grade inflammation is related to their capacity for modulating leukocyte and adipocyte function, thus reducing expression and production of inflammatory cytokines and chemokines [104]. Other studies have demonstrated that SCFAs decrease the LPS-induced neutrophil release of TNF-α. Accordingly, exposure of adipocytes to SCFAs significantly downregulates several inflammatory cytokines and chemokines, decreasing the risk of T2DM and CVD.

Another important function of SCFAs in T2DM is binding with GPCRs that lead to the activation of various biological effects. For example, SCFAs promote the secretion of Glucagon-like peptide-1 (GLP-1), a gut-derived incretin hormone that is essential for normal glucose tolerance. This hormone is primarily synthesized by intestinal L-cells and released into circulation during meal ingestion. GLP-1 suppresses glucagon secretion, decreases hepatic gluconeogenesis, improves insulin sensitivity, and enhances central satiety, resulting in weight loss. The therapeutic potential of GLP-1 has already been established, as several pharmaceutical agents promoting its effects are successfully used for blood glucose management in individuals with T2D, as well as for bodyweight management in obese individuals. Intravenous GLP-1 is highly effective in stimulating insulin secretion and reducing hyperglycemia in T2DM patients. Furthermore, GLP-1 mimetics and a dipeptidyl peptidase-4 inhibitor are commonly used therapeutic approaches in clinics [105]. In particular, injectable GLP-1 mimetics are associated with reductions in fasting and postprandial glucose concentrations, decreased hemoglobin A1c (HbA1c), and significant weight loss. 

Among SCFAs, butyrate seems to have a lower potency than acetate and propionate in stimulating GLP-1 secretion [106]. However, butyrate treatment of the human L-cell line has resulted in enhanced secretion of GLP-1 and increased expression of genes involved in GLP-1 synthesis and secretion. Furthermore, the beneficial effects of administration of the prebiotic showed the association with protection from body weight gain, decreased insulin resistance, and increased GLP-1 secretion [107]. Mechanisms by which butyrate increases GLP-1 secretion are still a matter of debate. The poor responsiveness of GLP-1 to SCFAs in GLUTag cell line has been associated with the very low expression of GPR43 [106]. Lower levels of insulin were observed in GPR43 knockout mice compared with wild-type controls after a prolonged period on a high-fat diet [108]. Conversely, another study suggests butyrate interaction on L-cells might be mediated by GPR41, since increases in the secretion of GLP-1 induced by butyrate were associated with increased GPR41 expression. Moreover, butyrate-induced total GLP-1 secretion was attenuated in the GPR41 knockout mice [108].

SCFAs are associated with regulation of insulin levels via GLP-1 expression, and result in improvement of metabolic functions in T2DM. These effects result from different tissues expressing SCFA receptors, and can thus respond to the beneficial effects induced by these molecules. The regulation of blood glucose concentrations exerted by SCFAs can occur through multiple mechanisms, including reduced insulin resistance from decreased inflammation, the contemporarily increased GLP-1 secretion that stimulates insulin release, and improved beta-cell function that contributes to the amelioration of glucose homeostasis.

### 3.2. Gut Microbiota and Inflammatory Bowel Disease (IBD)

Inflammatory bowel diseases (IBDs), including both Crohn’s disease (CD) and ulcerative colitis (UC), are chronic inflammatory conditions of the GI tract, resulting from altered interactions between gut microbiota and the intestinal immune system. These conditions result from a complex interplay among the host, gut microbiota, and environmental factors. Major shifts in gut microbial composition, including increases in facultative anaerobes and decreases in obligate anaerobic producers of SCFAs, commonly occur in IBD patients [109]. IBD involves a breakdown in relations between host immune responses and the microbial population resident in the GI tract. Although studies to date have failed to reveal a single etiological pathogenic species responsible for IBD, the current view is that while individual species may play significant roles in immunomodulation, collateral damage to the gut microbiota due to their loss or overabundance plays a key role in the persistence of inflammatory responses in chronic disease [97,110,111].

Evidence for the role of gut microbiota in IBD pathogenesis is provided through studies demonstrating that antibiotic use can reduce or prevent inflammation in both IBD patients and murine models of disease. Initial studies performed by Rutgeerts (1991) [112] and D’Haens (1998) [113] demonstrated the involvement of gut microbiota in IBD using clinical experiments that showed that diversion of the fecal stream improved symptoms of CD, and that postoperative exposure of the terminal ileum to luminal contents increased inflammation. In CD patients, increased gut permeability promotes bacterial translocation through the intestinal mucosa. In healthy individuals, the gut barrier consists of an intact layer of epithelial cells that are tightly connected by a surrounding system of tight junction strands from the claudin protein family. Gut barrier dysfunctions observed in patients with mild to moderately active CD result mostly from the up-regulation of claudin 2, and the down-regulation and redistribution of claudins 5 and 8, leading to discontinuous tight junctions [114,115]. In IBD patients, gut dysbiosis occurs due to increasing colonization of an enteric pathogen, host-mediated inflammatory responses, or a combination of the two. Pathogens can subvert the inflammatory response and take advantage of inflammation to breach the barrier imposed by commensal resident microbes and the intestinal mucosa itself. *Salmonella typhimurium* is one such pathogen that induces inflammation responses that alter the composition of the gut microbiota and promote its own growth in mouse models [116,117]. Another pathogen, *Citrobacter jejuni*, is among the most common causes of bacterial infection in humans. In mice, dextran sulfate sodium (DSS)-induced colitis non-inflammatory infection with *Citrobacter jejuni* has decreased the total number of colonic bacteria, but DSS-induced inflammation promoted overgrowth of *Enterobacteriaceae*. Both conditions were necessary to reach maximal gut dysbiosis [118,119]. Commensal gut bacteria can also alter pools of available metabolites, thereby modifying host-generated signaling molecules. Harnessing the ability of the microbiota to affect host immunity is considered an important therapeutic strategy for patients with IBD.

Notably, numerous studies have documented alterations in microbiota associated with CD, particularly reduced abundances of the phylum *Firmicutes* and a concomitant increase in *Proteobacteria* [114,120]. Specifically, *Clostridium* clusters IV and XIV have exhibited relatively lower abundance in IBD patients compared to healthy controls, suggesting their loss may deplete the microbiota of key anti-inflammatory metabolites or other cell-associated immunomodulatory ligands. Both CD and UC gut microbiota exhibit general decreases in taxonomic diversity relative to healthy controls, along with phylum-level decreases in *Firmicutes* and increases in *Proteobacteria*. The abundance of bacteria from the *Clostridia* family have also been altered, with decreases in *Roseburia* and *Faecalibacterium* and increases in *Ruminococcus* and the *Enterobacteriaceae* family in IBD patients [120,121]. Together, these findings suggest that IBD is ultimately linked to inflammation that may be largely associated with microbially derived or modified metabolites. 

Mutations of the NOD2 gene increase susceptibility to CD. PRRs, including NOD-like receptors (NLRs), TLRs, and others, play a crucial role in the innate immune response by recognizing pathogen-associated molecular patterns derived from a diverse collection of microbial pathogens. Evidence from IBD genetic studies have demonstrated that several innate immune genes have functionally relevant polymorphisms. Of those studied, NOD2 genetic variants conferred the greatest risk. NOD2 functions as an intracellular sensor for microbial pathogens and plays an important role in epithelial defense. NOD2, a member of the NLR family, functions as an important intracellular sensor for bacteria, detecting peptidoglycans through the recognition of muramyl dipeptide [121]. NOD2 activates nuclear factor NF-kB, which has an inhibitory role and acts as an intracellular receptor for components of microbial pathogens. After binding with intracellular muramyl dipeptide (MDP), NOD2 recruits the adaptor protein RIP2 to activate NF-κB and initiate a pro-inflammatory response. NOD2 was the first identified gene strongly associated with susceptibility to CD patients, and NOD2 loss of function mutations occur in IBD patients. In one study, the number of intestinal intraepithelial lymphocytes (IELs) were reduced significantly in NOD2−/− mice, and the residual IELs displayed reduced proliferation and increased apoptosis. NOD2 signaling maintained IEL abundances via recognition of gut microbiota and IL-15 production. Recognition of gut microbiota by NOD2 is important to maintain the homeostasis of IELs, providing a clue linking NOD2 variation to impaired innate immunity and higher susceptibility to CD [121].

### 3.3. Gut Microbiota and Systemic Lupus Erythematosus (SLE)

Systemic lupus erythematosus (SLE) is a prototypic autoimmune disease affecting multiple organs and is characterized by autoantibody production, reflecting dysregulation of both the innate and adaptive immune systems. Loss of tolerance to various self-antigens is a critical feature of SLE. While the pathogenesis is not fully understood, a combination of environmental factors, host genetic background, and gut microbiota can contribute to SLE [122]. The associated loss of immune tolerance can be caused by altered gut microbial composition, with both experimental and clinical models providing evidence of the strong relationship between gut microbiota and SLE [123,124,125]. 

Gut microbiota can stimulate an immune response against the host if the mechanisms of tolerance fail for any of several reasons. In normal cell apoptotic conditions, the host immune system does not involve the release of nuclear antigens. In SLE, however, external stimuli such as bacterial infection, toxins, or ultraviolet (UV) light induce DNA damage and keratinocyte apoptosis. The resulting prolonged autoantigen exposure increases stimulation of host immune cell responses [126]. SLE also induces T-cell activation via suppression of Treg cells in a type I interferon (IFN)-dependent manner. Enhanced levels of nucleic acid-containing cell fragments activate the type I IFN pathway through the action of nucleic acid recognition receptors TLR7 or TLR9 [127,128]. Type I IFNs and other cytokines promote B-cell maturation and survival, inducing the B-cell hyperactivity characteristic of SLE. B-cells produce high-affinity autoantibodies against self-antigens that cause inflammation and tissue damage. Autoantibodies and immune complexes (ICs) mediate inflammation and tissue damage by activating complement cascades and binding to Fc receptors on inflammatory cells. Fc receptors are a heterogeneous group of hematopoeitic cell surface glycoproteins that facilitate the efficiency of antibody–antigen interactions with effector cells of the immune system. These receptors are involved in a variety of humoral and cellular immune responses including phagocytosis, antibody-dependent cellular cytotoxicity, cytokine and chemokine expression, B-cell activation, and IC clearance. Many studies have shown that the level of expression and function of the Fc region in IgG (FcγR) are altered in SLE. Consistent with its role in the pathogenesis of SLE, FcγR stimulates the production of autoantibodies, resulting in inflammation and IC handling. Altered or delayed clearance of autoantibody containing ICs results in their deposition in various tissues, eliciting inflammation and damage by engaging FcγRs and complement cascades [129,130]. 

An increase in follicular Th cells and defects in Treg cells have also been implicated in SLE pathogenesis. Scalapino and colleagues (2006) [131] demonstrated in adoptive transfer experiments that Treg cells delay disease progression and reduce the mortality of lupus-prone mice. Treg cells are a subset of CD4+ T cells that maintain self-tolerance by suppressing autoreactive lymphocytes, leading to the hypothesis that defects in Treg cells contribute to SLE pathogenesis. Activated T cells release proinflammatory cytokines and induce B-cell secretion of autoantibodies. Thus, activation of both B cells and T cells induce innate and adaptive immune responses toward autoimmunity [36,132]. In germ-free murine models, T-cell subsets in the gut are abnormal, displaying reduced responses of Th17 in the lamina propria of the small intestine and Treg in the colonic lamina propria [133,134]. Gut microbiota has been associated with an imbalance in the proportions of Th17 and Treg cells in SLE patients [125]. In patients with SLE, gut microbial composition significantly enriches several genera, including *Klebsiella*, *Rhodococcus*, *Eggerthella*, *Eubacterium*, *Prevotella*, and *Flavonifractor*. In contrast, the *Firmicutes/Bacteroidetes* ratio and abundance of *Dialister* and *Pseudobutyrivibrio* have been decreased in SLE patients [123]. It remains unclear whether the observed changes in commensal bacteria are a consequence of the disease process or if gut dysbiosis contributes to SLE onset. However, manipulation of gut microbiota in murine models with antibiotic treatment provides further evidence of gut microbiota influences on systemic immune homeostasis [135,136]. 

Furthermore, studies show a significant reduction of *Lactobacillaceae* and an increase of *Lachnospiraceae* in female lupus-prone mice [137]. *Lactobacillus*, a genus in the *Lactobacillaceae* family, is a common resident microbiota in human GI tracts. Some *Lactobacillus* species are used as probiotics due to their anti-inflammatory properties. The relative lack of *Lactobacillus* spp. is most prominent before, but not after, disease onset. *Lactobacillus* treatment contributes to an anti-inflammatory environment by decreasing IL-6 and increasing IL-10 production in the gut. In circulation, *Lactobacillus* treatment increased IL-10 and decreased IgG2a, a major immune deposit in MRL/lpr mice. Thus, *Lactobacillus* may play a preventive role in lupus pathogenesis [138]. However, *Lactobacillus* has played an opposite role in alternative lupus murine models, where the relative abundance of *Lactobacillus* species tended to correlate positively with decreased renal function and higher-level systemic autoimmunity in NZB/W F1 mice [124,139]. Thus, exploration of the gut microbiota in murine and human lupus has afforded new insights into the role played by the bacteria in SLE.

Divergent associations were observed for pro- and anti-inflammatory free fatty acids (FFAs) with endothelial activation biomarkers in lupus patients, supporting the link between the gut microbiota and the host metabolism in the pathological framework of SLE [140]. The altered production of SCFAs related to the intestinal dysbiosis highlights the role for the gut microbiota in the maintenance of serum FFA levels, further pointing to SCFAs as potential orchestrators of the cross talk between gut microbiota and the host metabolism [140]. Additionally, five perturbed metabolic pathways were identified in feces of SLE patients, including aminoacyl-tRNA biosynthesis, thiamine metabolism, nitrogen metabolism, tryptophan metabolism, and cyanoamino acid metabolism [141]. The aforementioned metabolites could also be used as non-invasive biomarkers for SLE, as long as the effects of potential cofounders, such as medications, co-morbidities, smoking, and diet are considered [142]. Given the role of metabolites in autoimmune diseases, one proposed therapeutic strategy is to promote generation of SCFAs in the gut in order to impel the differentiation of naïve CD4+ T-cells into Treg cells, avoiding differentiation into Th1 and Th17 cells. One may support the growth and proliferation of SCFA-producing gut microbes by using appropriate types of dietary fiber [143], prebiotics such as fructo-oligosaccharides [144,145], or probiotics [146]. Oral delivery of SCFAs may be problematic as they are metabolized extensively in the upper portions of the intestinal tract [147].

## 4. Conclusions & Future Directions

Data from both human patients and mouse models suggest direct links, largely through immunological mechanisms, between gut microbiota and a number of common human illnesses. Many of these links are statistical associations without an understanding of causality, susceptibilities to genetic allelic components, and/or environmental factors that may act in tandem. Dependence upon murine models has both advanced and hindered the field, as many variables affecting mice are not adequately controlled. Further, mice are genetically inbred (lacking the complex genetic variability of humans), have an average life span of two years, and display different receptor sensitivities in comparison with humans. While human and mouse immune systems have many similarities, there are areas that are widely disparate. The murine and human gut microbiota share two major phyla, but 85% of their bacterial genera are not found in humans, and their microbiota is largely driven by a coprophagic diet different from common human diets. Differences in the immune system must also be considered, and the development of humanized immune systems in mouse models will advance our understandings. Therapeutics for altering the gut microbiota in animals is in its infancy, and more basic research is required. In total, these findings indicate that modifications on gut microbiota and their metabolism may be a viable therapeutic and diagnostic target of autoimmune diseases.

Despite our nascent understanding of the role gut microbiota play in the onset and progression of human disease, several mechanisms have been elucidated by which gut microbiota and their metabolic products interact with and regulate both innate and adaptive immune systems. Central to these interactions, and a common theme among seemingly disparate disease states, is subversion of the mucosal layer produced by intestinal epithelial cells and leakage of bacteria and metabolic products through the intestinal barrier. Translocation from the leakage activates a variety of signaling pathways in a manner dependent upon the location sink within the host and the invading species or metabolites. These pathways then stimulate inflammatory immune responses through the production of pro- and anti-inflammatory cytokines, and alterations to B- and T- cell populations. At each interaction, host genetics likely play a role through receptor activation rates and signaling intensity. The microbial production of SCFAs in the gut, particularly butyrate, reduces intestinal inflammation and increases the integrity of the intestinal barrier, minimizing leakage and bacterial translocation.

## Figures and Tables

**Figure 1 microorganisms-08-01587-f001:**
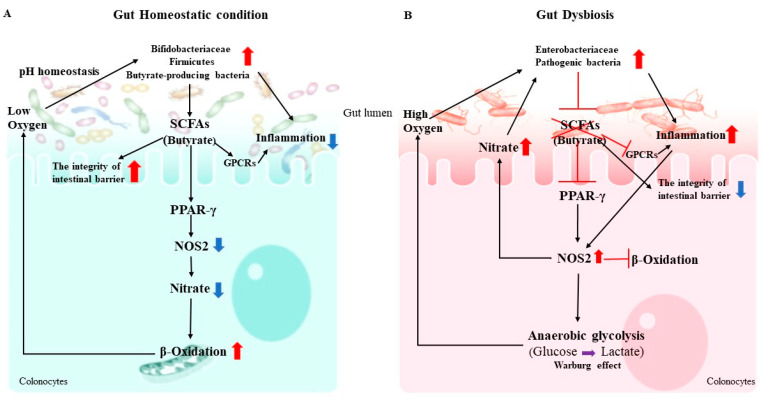
The interaction between microbiota-derived metabolism in the gut epithelium. (**A**) In the gut homeostatic condition, gut microbiota, especially butyrate-producing bacteria, converts fiber into fermentation products such as SCFAs. These SCFAs stimulate a PPAR-γ–dependent activation of mitochondrial-oxidation, thereby lowering epithelial oxygenation. SCFAs also directly bind G protein-coupled receptors (GPCRs), such as GPR41, GPR43, and GPR109A, on the surface of epithelial cells and immune cells that lead to decrease inflammation in the gut. Transport or diffusion of SCFAs into host cells results in their metabolism and/or inhibition of histone deacetylase (HDAC) activity (not shown). (**B**) During the gut dysbiosis, *Enterobacteriaceae* uses its virulence factors to trigger neutrophil transepithelial migration, which leads to a depletion of SCFA-producing bacteria, thereby lowering the luminal concentration of short-chain fatty acids such as butyrate. The consequent metabolic reprogramming of the epithelium increases the luminal bioavailability of oxygen (O2) and lactate. Arrows represent increases (red) and decreases (blue) in bacterial abundances, metabolites, and downstream effects observed in the homeostatic and dysbiotic guts.

**Figure 2 microorganisms-08-01587-f002:**
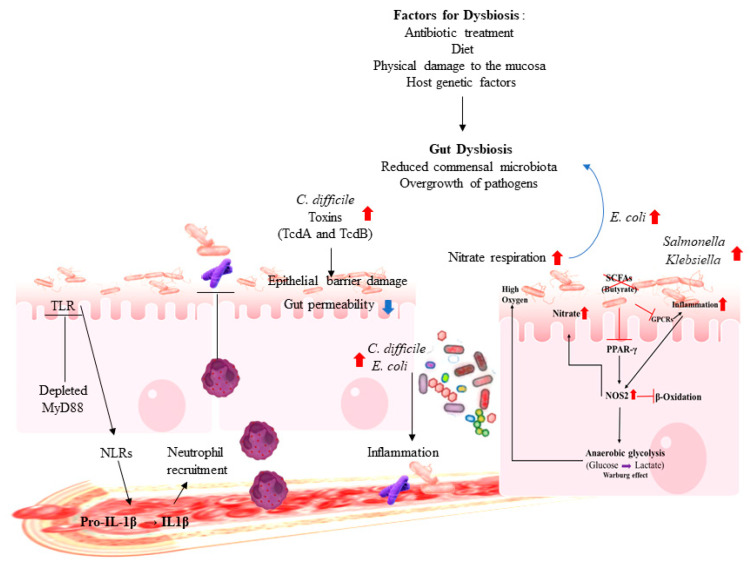
Gut dysbiosis. Antibiotic treatment or other factors can disrupt the commensal microbial community, resulting in diminished resistance to colonization by pathogens and the outgrowth of indigenous pathobionts, such as *Clostridium difficile*. *C. difficile* can produce toxins, like TcdA and TcdB, that destroy the epithelial barrier and increase gut permeability. This toxin-mediated epithelial damage can cause systemic circulation of bacteria, which is associated with increasing systemic inflammation. Pathogen-induced gut inflammation confers a growth advantage to the pathogen through the generation of molecules such as inducible nitric oxide synthase (iNOS) by host innate immune cells leading to the release of nitrate (NO_3_^−^), which can be used as an electron acceptor by *Escherichia coli* to generate energy through nitrate respiration. Pathogen infection results in the conversion of pro-IL-1β into the enzymatically active mature form of IL-1β, which promotes neutrophil recruitment and pathogen eradication. Bacterial toxins stimulate the NLRP3 inflammasome, driving the proteolytic activation of caspase-1, which results in the release of mature, biologically active IL-18 and IL-1β. Arrows represent increases (red) and decreases (blue) in bacterial abundances, metabolites and downstream effects observed in the antibiotic treated dysbiotic gut.

**Figure 3 microorganisms-08-01587-f003:**
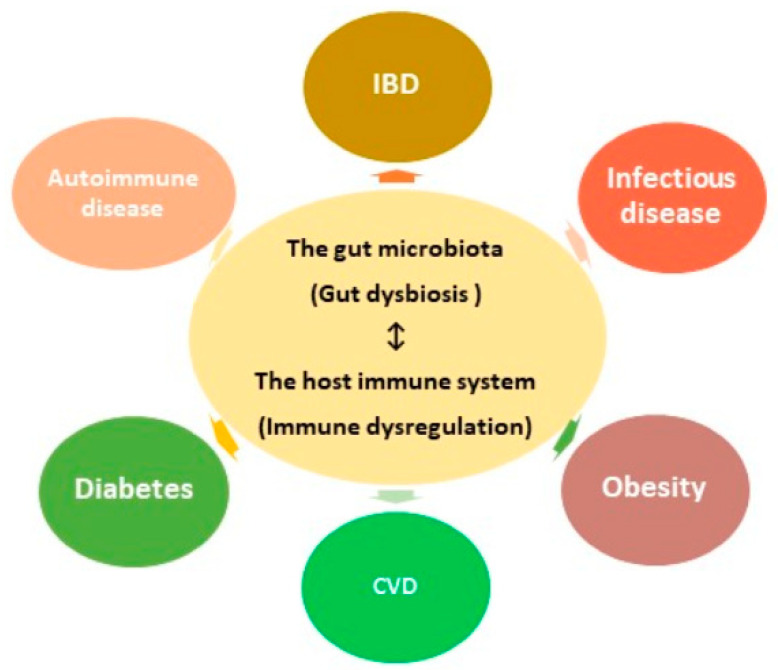
Human gut microbial dysbiosis has a close relationship with diseases. The immune system–gut microbiota crosstalk is of sublime importance in understanding the role of dysbiosis-driven diseases in humans. Gut dysbiosis induces immune dysregulation and subsequently increase the risk of developing diseases, including inflammatory bowel disease (IBD), diabetes, obesity, cardiovascular diseases (CVDs), infectious disease, and autoimmune disease.

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
