# Peer review of "Gut Microbiota and Immune System Interactions"

_microorganisms, 2020, doi:10.3390/microorganisms8101587_

Round 1
Reviewer 1 Report
To increase the readability the first section could be easily split into two sections: 1. Gut microbiota metabolites and 2. Immune regulation (of gut microbiota).
The second section "Gut Dysbiosis and Immune Dysregulation" is written in a slightly superficial way. It deserves more in-depth view on the problem.
The third part includes the role of microbiota in selected disorders. It is not clear why these disease have been selected and, e.g. obesity or liver failure has not. This should be somehow explained.
Major problem is the discrepancy between the title and the contents. Although the title is "Diversity and Dynamics of the Gut Microbiota and Immune Cells", there is not much information of critical insights on diversity, and even less on dynamics of gut microbiota and immune cells. I would recommend to change the title to better reflect the contents or to change the contents (add information on the dynamics) to fit the title.
Minor comments:
References in the paragraph starting on line 537 are missing square brackets.
Please check and unify the use of the terms microbiota/microbiome along the text.
Additional figure(s) explaining dysbiosis and immune dysregulation would be beneficial.
Author Response
Reviewer #1
Major comments
Comment 1: “To increase the readability the first section could be easily split into two sections: 1. Gut microbiota metabolites and 2. Immune regulation (of gut microbiota)”
Response 1: We split the first section into two parts, but joined the second part (immune regulation of gut microbiota) with the second section (Gut Dysbiosis and Immune Dysregulation.
Comment 2: The second section “Gut Dysbiosis and Immune Dysregulation” is written in a slightly superficial way. It deserves more in-depth view on the problem.
Response 2: We added more information on this section.
Comment 3: The third part includes the role of microbiota in selected disorders. It is not clear why these disease have been selected and, e.g. obesity or liver failure has not. This should be somehow explained.
Response 3: We included an introductory paragraph for the third part that provides a rationale for the diseases discussed.
Comment 4: Major problem is the discrepancy between the title and the contents. … I would recommend to change the title to better reflect the contents or to change the contents to fit the title.
Response 4: We altered the title to remove the references to diversity and dynamics.
Minor comments
Comment 1: References in the paragraph starting on line 537 are missing square brackets.
Response 1: We removed references from the concluding paragraph. Note, references are still included earlier in the manuscript.
Comment 2: Please check and unify the use of the terms microbiota/microbiome along the text.
Response 2: We have standardized to the term microbiota.
Comment 3: Additional figure(s) explaining dysbiosis and immune dysregulation would be beneficial.
Response 3: Gut dysbiosis figure has been added.
Reviewer #2
Comment 1: Well written review on a very interesting topic. Despite being well written there are sometimes strange typos which should be checked. Some examples: (lines 25, 27/28, 163, 185)
Response 1: We have corrected the typos identified and have re-read the manuscript to identify and correct further typographical errors.
Reviewer #3
Comment 1: In conclusion, I can state that the paper summarizes the current state of knowledge on the correlation of intestinal microbiota and immune system in a modern and interesting way. In my opinion, the paper can be accepted for publication without corrections.
Response 1: Thank you so much for your kind words.
Reviewer 2 Report
Well written review on a very interesting topic. Despite being well written ther aer sometimes strange typo's which should be checked. Some examples:
- line 25: musical instead of mucosal
- line 27/28: pivotal role of microbiota.... play
- line 163: microbiota where it should be microbe
- line 185: IgA-producing cells IgA?
Author Response
Reviewer #1
Major comments
Comment 1: “To increase the readability the first section could be easily split into two sections: 1. Gut microbiota metabolites and 2. Immune regulation (of gut microbiota)”
Response 1: We split the first section into two parts, but joined the second part (immune regulation of gut microbiota) with the second section (Gut Dysbiosis and Immune Dysregulation.
Comment 2: The second section “Gut Dysbiosis and Immune Dysregulation” is written in a slightly superficial way. It deserves more in-depth view on the problem.
Response 2: We added more information on this section.
Comment 3: The third part includes the role of microbiota in selected disorders. It is not clear why these disease have been selected and, e.g. obesity or liver failure has not. This should be somehow explained.
Response 3: We included an introductory paragraph for the third part that provides a rationale for the diseases discussed.
Comment 4: Major problem is the discrepancy between the title and the contents. … I would recommend to change the title to better reflect the contents or to change the contents to fit the title.
Response 4: We altered the title to remove the references to diversity and dynamics.
Minor comments
Comment 1: References in the paragraph starting on line 537 are missing square brackets.
Response 1: We removed references from the concluding paragraph. Note, references are still included earlier in the manuscript.
Comment 2: Please check and unify the use of the terms microbiota/microbiome along the text.
Response 2: We have standardized to the term microbiota.
Comment 3: Additional figure(s) explaining dysbiosis and immune dysregulation would be beneficial.
Response 3: Gut dysbiosis figure has been added.
Reviewer #2
Comment 1: Well written review on a very interesting topic. Despite being well written there are sometimes strange typos which should be checked. Some examples: (lines 25, 27/28, 163, 185)
Response 1: We have corrected the typos identified and have re-read the manuscript to identify and correct further typographical errors.
Reviewer 3 Report
In the paper "Diversity and Dynamics of the Gut Microbiota and
3 Immune Cells" authors have undertaken to describe the relationship between intestinal microbiota and the immune system.
The work consists of 21 pages of text (including bibliography and two figures). The issues related to the interaction between the immune system and microorganisms in the gastrointestinal tract are discussed in a modern and comprehensive way. The discussed problems are presented at the cellular and molecular level.
The bibliography of the paper consists of 147 scientific articles. It should be emphasized that the quoted papers present the current state of knowledge on the topics presented in the paper. In conclusion, I can state that the paper summarizes the current state of knowledge on the correlation of intestinal microbiota and immune system in a modern and interesting way.
In my opinion, the paper can be accepted for publication without corrections.
Author Response

(The authors gave the same response as above.)

Round 2
Reviewer 1 Report
The comments have been properly addressed.